# Dietary Micronutrient Supplementation for 12 Days in Obese Male Mice Restores Sperm Oxidative Stress

**DOI:** 10.3390/nu11092196

**Published:** 2019-09-12

**Authors:** Nicole O. McPherson, Helana Shehadeh, Tod Fullston, Deirdre L. Zander-Fox, Michelle Lane

**Affiliations:** 1Robinson Research Institute, School of Medicine, University of Adelaide, Adelaide 5005, Australia; hshehadeh@repromed.com.au (H.S.); tod.fullston@repromed.com.au (T.F.); mlane@monashivfgroup.com (M.L.); 2Freemasons Foundation Centre for Men’s Health, University of Adelaide, Adelaide 5005, Australia; 3Repromed, Dulwich 5065, Australia; dzander@monashivfgroup.com; 4Monash IVF Group, Melbourne 3000, Australia; 5Future Industries Institute, University of South Australia, Adelaide 5095, Australia

**Keywords:** DNA damage, epididymis, obesity, sperm maturation, sperm function, dietary supplementation, antioxidant, micronutrient, reactive oxygen species

## Abstract

Male obesity, which often co-presents with micronutrient deficiencies, is associated with sub-fertility. Here we investigate whether short-term dietary supplementation of micronutrients (zinc, selenium, lycopene, vitamins E and C, folic acid, and green tea extract) to obese mice for 12 days (designed to span the epididymal transit) could improve sperm quality and fetal outcomes. Five-week-old C57BL6 males were fed a control diet (CD, *n* = 24) or high fat diet (HFD, *n* = 24) for 10 weeks before allocation to the 12-day intervention of maintaining their original diets (CD, *n* = 12, HFD *n* = 12) or with micronutrient supplementation (CD + S, *n* = 12, HFD + S, *n* = 12). Measures of sperm quality (motility, morphology, capacitation, binding), sperm oxidative stress (DCFDA, MSR, and 8OHdG), early embryo development (2-cell cleavage, 8OHdG), and fetal outcomes were assessed. HFD + S males had reduced sperm intracellular reactive oxygen species (ROS) concentrations and 8OHdG lesions, which resulted in reduced 8OHdG lesions in the male pronucleus, increased 2-cell cleavage rates, and partial restoration of fetal weight similar to controls. Sub-fertility associated with male obesity may be restored with very short-term micronutrient supplementation that targets the timing of the transit of sperm through the epididymis, which is the developmental window where sperm are the most susceptible to oxidative damage.

## 1. Introduction

The prevalence of obesity has more than doubled in the last three decades making obesity in adults more common than under-nutrition [1]. This is most evident in westernized societies including Australia [2], the United States, the United Kingdom, and parts of Europe [3], whereby approximately 30% of adult men are now classified as obese (body mass index—BMI > 30 kg/m^2^). Besides the recognized increased risk of developing chronic diseases including type II diabetes, heart disease, and some cancers, obesity is also associated with sub-fertility [4]. A recent systematic review found that men who are obese have decreased total sperm counts, lower sperm mitochondrial membrane potential, and increased sperm DNA damage, resulting in a higher odds ratio of experiencing infertility in the general population [5]. These sperm traits were coupled with reduced live birth rates after assisted reproductive treatment [5]. Alarmingly, male obesity at conception can also impact the next generation, manifesting as altered birth weights in humans [6,7] and animal models [8,9,10,11,12,13], while also making offspring more susceptibility to metabolic syndrome [8,9,14], sub-fertility [15], fatty liver disease [16], kidney disease [17], and hypertension [18]. Therefore, there is a need for clinically applicable interventions in obese men prior to conception to help improve both sperm quality but to also break the transgenerational disease cycle. 

A primary hypothesis for how male obesity reduces sperm quality, pregnancy rates, and perturbs offspring health is through oxidative stress in sperm [19,20,21]. A large proportion of sperm damage because of increasing concentrations of reactive oxygen species (ROS) occurs during the epididymal transit, where sperm spend ~9.5 days in mice [22] and 2–4 days in humans [23] completing their final stages of maturation [24]. This is because the epididymis contains a semi-permeable barrier exposing the developing sperm to the external environment [25] and as mature sperm lack the intracellular antioxidant enzyme protection, because of cytoplasmic shedding during spermiogenesis in the testes, this makes them highly susceptible to oxidative damage during this transit period [21]. Therefore, directly targeting ROS production in obese men during this critical epididymal window may reduce sperm ROS concentrations and therefore improve fetal outcomes.

We have previously shown that lifestyle interventions including changing to a low saturated fat nutrient matched diet coupled with light exercise for 2.5 months in our rodent model of obesity reduces adiposity, reduces sperm ROS, and associated oxidative DNA damage, and reverses the adverse fetal outcomes associated with male obesity [11,26,27,28]. However, issues with compliance and remission especially in humans plague long-term diet and exercise interventions as suitable viable clinical options [29], especially in patients undergoing assisted reproductive technologies (ART). Therefore, shorter clinical interventions to generate improvements to fertility are desirable. 

It is well-known that nutritional intake is directly related to an organism’s oxidative load [30], with diets favoring seafood, poultry, whole grains, fruits, low-fat dairy, skimmed milk, fruits and vegetables related to better semen quality and reduced sperm oxidative damage in men [31,32,33]. Obese individuals often present with micronutrient deficiencies in a number of vitamins and minerals that have antioxidant properties including but not limited to vitamin E (alpha-tocopherol), vitamin C (ascorbic acid), beta-carotene, vitamin A, folic acid, B vitamins, and selenium [34,35], with a number of these molecules shown to reduce oxidative damage in sperm of sub-fertile patients when taken orally [36,37]. Therefore, improving micronutrient deficiency in obese men through oral supplementation may be a viable intervention for reducing sperm ROS concentrations and improving fetal outcomes. Indeed, studies in animal models have shown that long term antioxidant and/or micronutrient supplementation (3 months) provided to high fat diet fed obese rodents was able to restore basic sperm function (count and motility) [38] and reduce oxidative testicular injury [39,40]. However, whether a shorter intervention can elicit an improvement in sperm quality is unknown. 

Utilizing our well-established mouse model of diet induced male obesity and sub-fertility [9,10,15,26,41], we hypothesized that a short-term (12 days) intervention to approximate the epididymal transit of oral micronutrient supplementation can restore sperm oxidative damage and fetal outcomes. 

## 2. Materials and Methods 

### 2.1. Animals, Diet, and Micronutrient Supplementation

Five- to six-week-old male C57BL6 mice (*n* = 48) were randomly assigned in the pre-intervention period to either a control diet (CD) (*n* = 24) or a high fat diet (HFD) (*n* = 24) (SF12-012 and SF13-109 respectively; Specialty Feeds, Perth, Australia; Table 1) for 10 weeks. We have previously shown that this time on the diet initiates increased adiposity and perturbed sperm function in those animals allocated to the HFD [9,15,41]. After this initial diet phase mice were allocated to one of four diets for the intervention period, whereby mice initially fed a CD either (i) continued a CD, or were fed a (ii) CD supplemented with micronutrients (CD + S; SF12-013; Specialty Feeds, Table 1), and mice initially fed a HFD either (iii) continued a HFD or were fed a (iv) HFD supplemented micronutrients (HFD + S; SF13-110; Specialty Feeds, Table 1) for an additional 12 days. We added micronutrients to the diets that are known to be reduced in obese individuals [34,35], are important for sperm function [42,43,44,45] and have been shown to improve sperm oxidative damage after oral administration in sub fertile men [37]. The 12-day duration was used to cover the duration of epididymal transit by sperm in mice (~9.5 days) [22]. Animals were housed individually with ad libitum access to food and water. Individual body weights were recorded weekly. The use and care of all mice were approved by the Animal Ethics Committee of the University of Adelaide (M-165-13) and were handled in accordance with the Australian Code of Practices for the Care and Use of Animals for Scientific Purposes.

### 2.2. Glucose and Insulin Tolerance Testing

An intraperitoneal (IP) glucose tolerance test (GTT) and an IP insulin tolerance test (ITT) were performed 9–10 weeks during the pre-intervention phase and again after 12 days of micronutrient supplementation. The GTT was conducted after 6 h of fasting by IP injection of 2 g/kg 25% glucose solution (Sigma-Aldrich, St. Louis, Miossouri, USA) and the ITT was conducted by IP injection of 1.0 IU/kg insulin on fed mice (Actapid; Novo Nordisk, Bagsvaerd, Denmark). Tail blood glucose concentrations were measured using a glucometer (HemoCue, Angelholm, Sweden) at 0 (pre-injection), and 15, 30, 60, and 120 min post-injection. Data were expressed as means of blood glucose concentration per group and area under the curve (AUC, min.mol) calculated for GTT or area above the curve (AAC, min.mol) calculated for ITTs.

### 2.3. Body Composition and Serum Metabolites and Hormones

Individual body weights were recorded weekly during the pre-intervention and the intervention period. After the 12-day intervention period (~16.5–17.5 weeks old), males were fasted overnight and blood plasma (~400 µL) was collected at post mortem by a cardiac puncture under anesthesia of 5% Avertin (2-2-2 Tribromethanol, Sigma Aldrich, St Louis, Missouri, USA). Serum glucose, cholesterol, high-density lipoproteins (HDL), triglycerides, and non-esterified free fatty acids (NEFA) concentrations were assessed using a COBAS Mira automated sample system (Roche Diagnostics, Basel, Switzerland). Serum testosterone concentrations were measured using a commercially available ELISA kit (R&D Systems, Minneapolis, MN, USA) as per the manufacturer’s instructions. In addition renal fat, dorsal fat, omental fat, gonadal fat, testes, seminal vesicles, liver, kidneys, spleen, and pancreas were collected and weighed at post mortem.

### 2.4. Collection of Mouse Sperm

Post mortem sperm was collected from the vas deferens and caudal epididymis and transferred into 1 mL of G-IVF + 10% HSA (Vitrolife, Goteberg, Sweden) before incubation at 37 °C for 10 min. All sperm assessments were conducted while blinded to the dietary groups.

### 2.5. Sperm Motility and Morphology

Sperm motility was determined manually under 40× magnification by counting duplicate samples of 200 sperm for motile, non-progressive motile, and immotile sperm. Sperm motility was expressed as a percentage of sperm in a given sample that were motile and non-progressive motile in accordance with the World Health Organization [46]. Sperm morphology was assessed using samples fixed in 3:1 methanol to acetone and subsequently stained with hematoxylin and eosin. A total of 200 individual sperm per mouse were assessed and classified as normal, or having an abnormal tail or heads as previously reported [26].

### 2.6. Sperm Capacitation

Sperm capacitation and acrosome reaction were measured using *Arachis hypogaea* (peanut) agglutinin (Lectin PNA; Molecular Probes, Eugene, USA) as previously described [47]. Sperm were incubated in G-IVF + 10% HSA for 1 h in 6% CO_2_ and 5% O_2_ at 37 °C, washed in PBS, and incubated in Lectin PNA Alexa 594 antibody (1:100) in PBS for 45 min. Samples were stained with Hoechst to identify sperm nuclei. A minimum of 200 sperm were counted per sample and were classified as capacitated, non-capacitated, or acrosome reacted.

### 2.7. Sperm Binding to the Zona Pellucida of MII Oocyte and Fertilisation Rates

To assess sperm binding to the zona pellucida, mature cumulus-enclosed oocytes (COCs) were collected from 4–5-week old CBAF1 female mice 12–13 h following ovulation induction by an IP injection of pregnant mare’s serum gonadotrophin (PMSG; Folligon; Intervet, Bendigo East, Australia) and hCG (Pregnyl; Organon, Australia) administered 48 h apart [48]. COCs were placed in 80 μL drops of G-IVF + 10% HSA in 6% CO_2_ and 5% O_2_ at 37 °C. After a 1-h incubation of the sperm, COCs were inseminated with 1 × 10^4^/mL of motile sperm and co-incubated at 6% CO_2_ and 5% O_2_ at 37 °C for 4 h. At 4 h post-insemination, sperm binding was assessed by counting the number of sperm bound to the zona pellucida of each MII oocyte as previously described [26]. Following this, zygotes were transferred to G1.3 media (Vitrolife, Goteberg, Sweden) and progression to the 2-cell embryo was assessed 24 h post co-incubation of sperm and eggs. Fertilization rates were assessed as the total number of COCs that cleaved to the 2-cell embryo.

### 2.8. Sperm Intracellular ROS Concentrations (DCFDA)

Intracellular reactive oxygen species (ROS) concentrations were assessed on progressively motile spermatozoa as per Bakos et al. [41]. Briefly, motile sperm were incubated with 5 µM DCFDA (2′,7′-dichlorodihydrofluoresce in diacetate; DCFDA; Sigma, Lenexa, USA) for 20 min at 37 °C, washed twice in GMOPS (Vitrolife, Goteberg, Sweden), and examined using a photometer attachment on a fluorescent microscope to derive a fluorescence reading for individually imaged sperm. A minimum of 20 motile sperm were measured per animal and expressed as relative fluorescent units. 

### 2.9. Sperm Mitochondrial ROS Concentrations (Superoxide-MitoSOX Red)

The intracellular generation of mitochondrial ROS was determined using MitoSox Red (MSR; Molecular Probes, Eugene, USA) as per Koppers et al. [49]. Sperms (10^6^/mL) were incubated with 0.05 µM of MSR and 2 µM of SytoxGreen (vitality stain) for 30 min at 37 °C, 6% CO_2_, and 5% O_2_. A negative control where sperm were only incubated in SytoxGreen was included. MSR and SytoxGreen fluorescence was measured on a FACSCanto flow cytometer (BD Bioscience, North Ryde, Australia). Non-specific sperm events were gated out and 20,000 cells were examined per sample. Results were expressed as percentage of live sperm positive for MSR. 

### 2.10. Sperm Oxidative DNA Damage (8OHdG)

Oxidative DNA damage was assessed in sperm as per McPherson et al. [50]. Briefly, sperm were fixed on slides in 3:1 methanol to acetone (Sigma Aldrich, St Louis, MO, USA) for 10 min. Sperm were permeabilized on the slide in 0.5% Triton X-100 (Sigma Aldrich, St Louis, MO, USA) and washed in phosphate buffered saline (PBS, Sigma Aldrich, St Louis, MO, USA). Sperm slides were then incubated in decondensing buffer (1 M HCl, 10 mM Tris buffer of 5 mM dithioretiol, Sigma Aldrich, St Louis, MO, USA) at 37 °C for 60 min. At room temperature, sperm DNA was then denatured (6 M HCL, 0.1% Triton X-100 in H_2_O, Sigma Aldrich, St Louis, MO, USA) for 45 min and then neutralized (100 µM Tris HCl in H_2_O, Sigma Aldrich, St Louis, MO, USA) for 20 min. Sperm slides were washed in PBS and then incubated overnight at 4 °C in 1:100 mouse anti-8-hydroxyguanosine (8OHdG) antibody (Abcam, Cambridge, UK) in 10% donkey serum whereas the negative control was incubated in 1:100 mouse serum instead of the anti-8OHdG antibody. On the second day, sperm slides were washed in PBS and incubated in 1:100 biotin-SP-conjugated AffiniPure donkey anti-mouse IgG (Jackson ImmunoResearch, Baltimore, PA, USA) at room temperature for 2 h. Sperm slides were then washed in PBS and incubated in 1:100 Cy3-conjugated streptavidin (Jackson ImmunoResearch) at room temperature for 1.5 h followed by a nuclear counter stain with Hoechst. Finally, sperm slides were loaded in ProLong Gold and glycerol solution (2 drops of ProLong Gold in 0.5mL glycerol; ProLong Gold, Molecular Probes, Eugene, USA; glycerol, Sigma-Aldrich, St Louis, MO, USA) and imaged under fluorescence microscopy. Using ImageJ (Version 1.48, National Institutes of Health, Bethesda, MD, USA), 30 sperm per mouse were assessed by quantifying 8OHdG fluorescence using ImageJ software. Results were expressed as a mean of fluorescence for 8OHdG (minus background) and then normalized to fluorescence from microspheres (Molecular Probes, Eugene, Oregon, USA). 

### 2.11. Oxidative DNA Damage (8OhdG) in the Pronuclear Embryo

Following fixation in 4% paraformaldehyde, pronuclear-staged embryos (PNs) at 18–19 hours post hCG injection/4–5 hours post fertilization ~PN3, were stored in PBS containing 3 mg/mL PVP (Sigma-Aldrich, St Louis, MO, USA) at 4 °C until an OxyDNA (Merck Millipore, Kenilworth, NJ, USA) test was conducted. Pronuclear-staged embryos were permeabilized in 0.25% Triton-X in PBS for 10 min at room temperature followed by incubation in 1:100 OxyDNA reagent for 1 h at 37 °C (as per the manufacturer’s instructions) [51]. PNs were counterstained with Hoescht (Sigma-Aldrich) for 5 min, mounted on glass slides and imaged using a confocal microscope. Z-stacks of 10 μm sections were collected of each pronucleus from each embryo (1024 × 1024 pixel size). Using ImageJ (Version 1.48, National Institutes of Health, Bethesda, MD, USA), confocal image stacks were reconstructed and regions of interest (ROI) were defined around both the maternal and the paternal pronucleus for fluorescence intensity quantification reflecting 8OHdG levels for each pronucleus. The size of the ROI (pronucleus) was also computed by ImageJ and the smaller of the two pronuclei for each embryo was determined as the maternal pronucleus and confirmed with its closer proximity to the polar body than the paternal pronucleus [52,53]. Paternal pronucleus 8OHdG fluorescence intensity quantification was normalized to the maternal pronucleus for each embryo and expressed as an average for each diet group.

### 2.12. Pregnancy

After the intervention period males had the opportunity to mate with two naturally cycling normal weight 10-week-old C57BL6 females for a period of 4 days. Mating was assessed by the presence of a vaginal plug the following morning and female mice were maintained on an ad libitum standard chow diet until day 18 of pregnancy. On day 18 of pregnancy, mothers were killed by cervical dislocation. The number of fetuses and resorption sites were recorded. Fetuses were dissected and removed of connective tissue and umbilical cords, weighed, and crown rump length was measured. Placentas were dissected and removed of connective tissue and weighed.

### 2.13. Statistical Analysis

All data were expressed as mean ± standard error of the mean (SEM) and checked for normality using the Kolmogorov–Smirnov test and equal variance using a Levene’s test. Statistical analysis was performed in SPSS (SPSS Version 18, SPSS Inc., Chicago, IL, USA) with AUC and AAC calculated in GraphPad Prism (GraphPad Software v6, San Diego, USA). A *p*-value <0.05 was considered significant and the statistical analysis accepted if power of the model was ≥80%. Body composition, hormones, and metabolites, GTT and ITT, all sperm measures and embryo oxidative DNA damage were analyzed by a one way ANOVA. Founder weekly weights were analyzed by a repeated measures general linear model with LSD post hoc test. Proportional embryo data were analyzed by a binomial generalized linear model (GLM) with LSD post hoc test. Fetal and placental outcomes were also analyzed by a GLM with LSD post hoc test with mother, father, and litter size fitted as fixed effect to control for fetuses born from the same mothers and same fathers and the size variation because of litter size. 

## 3. Results

### 3.1. Short-Term Micronutrient Supplementation Has No Effect on Body Composition or Serum Metabolites Outside of the Original Diets

#### 3.1.1. Pre-Intervention

Males fed a HFD were heavier from 7 weeks on the diet, which resulted in a 24% increase in total weight gained during the pre-intervention period compared with males fed a CD (Figure 1A,B, *p* < 0.05). Males fed a HFD were glucose intolerant as evident by a larger AUC and increased whole blood glucose concentrations at 30 and 60 min post glucose bolus during a GTT compared with males fed a CD (Figure 1C,D, *p* < 0.05). However, they displayed no signs of insulin resistance (Figure 1E,F).

#### 3.1.2. Post-Intervention

Males fed a HFD, irrespective of micronutrient supplementation (HFD and HFD + S), maintained their increased weights and glucose intolerance compared with males fed a CD with or without micronutrient supplementation (CD and CD + S) (Figure 2A–D, *p* < 0.05). Males fed a HFD irrespective of micronutrient supplementation also had increased fat mass (peri-renal, retroperitoneal, omental, dorsal, and gonadal) expressed as percentage of body weight, seminal vesicles weights (g), serum cholesterol, serum HDL, and serum triglycerides compared with males fed a CD with or without micronutrient supplementation (Table 2, *p* < 0.05). Interestingly, males fed a HFD with micronutrient supplementation had increased liver weights (g) compared to males fed a CD with micronutrient supplementation (Table 2, *p* < 0.05), with no change compared with the other diet groups. There was no effect of diet and/or micronutrient supplementation on any other body physiology measured (insulin tolerance, testes, pancreas and kidney weights, fasting serum glucose, NEFA, and testosterone concentrations) (Table 2 and Figure 2E,F). 

### 3.2. Short-Term Micronutrient Supplementation Improves Aspects of Sperm Function in CD-Fed Mice

Males fed a HFD without micronutrient supplementation had a reduced proportion of morphologically normal sperm, with increased abnormal tails and mid piece defects compared with males fed a CD (Table 3, *p* < 0.05). Micronutrient supplementation to males fed a CD improved proportions of normal morphological sperm with reduced proportions of sperm with abnormal tails and mid piece defects compared with all other diet groups (Table 3, *p* < 0.05). However, the same improvements to sperm morphology were not seen in males fed a HFD with micronutrient supplementation, with proportions of morphological normal sperm no different to males fed either a CD or HFD (Table 3) but reduced compared with males fed a CD plus micronutrient supplementation (Table 3, *p* < 0.05). This seemed to be due to increased tail defects of mice fed a HFD irrespective of micronutrient supplementation (HFD and HFD + S) compared with males fed at CD (CD and CD + S) (Table 3, *p* < 0.05). Males fed a HFD had a reduced proportion of sperm that had undergone capacitation, concomitant with an increased proportion of sperm that had spontaneously acrosome reacted compared with males fed a CD (Table 3, *p* < 0.05). Micronutrient supplementation had no effect on sperm capacitation irrespective of the original diet (Table 3). There was no effect of diet or micronutrient supplementation on sperm motility or the ability of sperm to bind to the zona pellucida of an MII oocyte. 

### 3.3. Short-Term Micronutrient Supplementation Reduces Intracellular Sperm ROS Concentrations and Oxidative DNA Damage in Males Fed a HFD

Males fed a HFD without micronutrient supplementation had increased intracellular (DCFDA) and superoxide (MSR) sperm ROS concentrations and oxidative sperm DNA damage (8OHdG) compared with males fed a CD (Figure 3A–C, *p* < 0.05). Micronutrient supplementation of males fed a HFD for 12 days reduced the sperm intracellular ROS concentrations and oxidative sperm DNA damage compared with males fed only a HFD (Figure 3A,C, *p* < 0.05) with concentrations similar to males fed a CD. However, the increased superoxide (MSR) concentrations generated by a HFD was not affected by micronutrient supplementation (Figure 3B). There was no impact of micronutrient supplementation on sperm ROS or oxidative DNA damage in males fed a CD (Figure 3A–C). 

### 3.4. Short-Term Micronutrient Supplementation Reduces Oxidative DNA Damage in the Early Pronuclear Embryo and Increases 2-Cell Cleavage Rates from Males Fed a HFD

8OHdG fluorescence in paternal pronuclei (PN) was quantified to determine if the increased oxidative DNA damage (8OHdG) found in sperm of HFD males persisted during very early fertilization events (18–19 hours post hCG injection/4–5 hours post fertilization ~PN3). Males fed a HFD produced early PN embryos with increased 8OHdG fluorescence in the paternal pronucleus compared with PN embryos produced from males fed a CD with or without micronutrient supplementation (Figure 4A, *p* < 0.05). Micronutrient supplementation of males fed a HFD reduced oxidative DNA damage in the male pronuclei compared with males fed only a HFD (Figure 4A, *p* < 0.05), with levels comparable to PNs CD embryos with or without micronutrient supplementation (Figure 4A). Again there was no effect of micronutrient supplementation in males fed a CD (Figure 4A).

Males fed a HFD without micronutrient supplementation had reduced percentage of embryos that successfully cleaved the 2-cell stage compared with males fed a CD with or without micronutrient supplementation (Figure 4B, *p* < 0.05). Micronutrient supplementation of males fed a HFD increased the percentage of 2-cell embryos compared with males fed only a HFD (Figure 4B, *p* < 0.05). However levels were still reduced compared with males fed a CD with or without micronutrient supplementation (Figure 4B, *p* < 0.05).

### 3.5. Short-Term Micronutrient Supplementation in Fathers Fed a HFD Partially Restored Fetal Weights Similar to Fathers Fed a CD

Males were mated with normal weight females to determine if the improvements to sperm ROS concentrations from 12 day micronutrient supplementation of HFD males could restore the abnormal fetal growth associated with HFD feeding [8,13,28]. There was no effect of diet and/or micronutrient supplementation on litter size or fetal lengths (mm) (Table 4, *p* > 0.05). Males fed a HFD without micronutrient supplementation had increased fetal weights compared with males fed only a CD (Table 4, *p* < 0.05). Micronutrient supplementation of either diet (CD/HFD) resulted in fetal weights similar to both males fed a CD and males fed a HFD (Table 4, *p* > 0.05). Feeding of a HFD and/or micronutrient supplementation to males reduced placental weights compared with males fed only a CD (Table 4, *p* < 0.05). This resulted in an increased fetal:placental weight ratio in males fed either a HFD and/or micronutrient supplementation compared with males fed only a CD (Table 4, *p* < 0.05).

## 4. Discussion

Increasing rates of male obesity continues to be a contributing factor to the worldwide decline in male fertility [54]. Diet and exercise interventions in both humans and animal models have shown promising results in reversing sub-fertility and adverse fetal outcomes associated with male obesity [11,26,27,28,33,55]. However, issues with compliance and remission frequently render lifestyle interventions ineffective as clinical interventions [29]. Therefore, there is a desire for interventions that have both high compliance and can be implemented into clinical ART practice. Here we show that as little as 12 days of micronutrient supplementation of male mice fed a high fat diet to induce obesity can restore sperm oxidative stress, fertilization rates and partially normalize fetal weights. 

One of the primary hypothesis as to how male obesity reduces sperm quality and perturbs offspring health is through oxidative stress of sperm [19,20,21]. Sperm acquire increased concentrations of ROS via either through the depletion of their own antioxidant protection, making them reliant on the antioxidant protection of the male reproductive tract, or exposure to excessive intrinsic or extrinsic ROS production [21]. Both of these have been shown to occur in male obesity, with (i) increased systemic oxidative stress and depletion of antioxidants [34] leading to reduced ROS mediated antioxidant capabilities of the testes and epididymis ultimately increasing the sperm ROS generation and DNA damage [56,57] and (ii) sperm from obese males, both in animal models and men, have been shown to produce more ROS in vitro [26,41,58]. A large proportion of sperm damage because of increasing concentrations of ROS occurs during the epididymal transit, where sperm spend ~9.5 days in mice [22] completing their final stages of maturation [24]. This 12-day micronutrient supplementation intervention was designed to target this critical final stage of sperm maturation where the sperm are most vulnerable to oxidative attack having shed their cytoplasmic defenses in the testes. 

One of the key findings from the manuscript was that a short-term intervention of 12 days of micronutrient supplementation reduced sperm intracellular ROS concentrations and associated oxidative DNA damage in obese males. The epididymis transfers a complex array of antioxidants including vitamin C, uric acid, taurine, and thioredoxin to sperm [24]. A number of the micronutrients that were chosen for inclusion in our study act or have antioxidant properties (these include vitamin E (alpha-tocopherol), vitamin C (ascorbic acid), lycopene, and selenium, zinc and green tea extract). In particular vitamin C and vitamin E are of interest because the body cannot synthesize them and therefore systemic concentrations directly relate to dietary intake [59,60]. Previous increased dietary intake of vitamin C in men reduced sperm oxidative DNA damage (8OHdG) [61], while *in vitro* exposure of sperm to 300 µM of ascorbic acid (the active component of vitamin C) had beneficial effects on sperm intracellular ROS levels and sperm chromatin integrity [62]. The same has also been shown for vitamin E both in vitro for protecting human sperm from induced oxidative stress at 40 μmol 1-1 vitamin E [63], and through oral supplementation of sub-fertile men to reduce sperm DNA fragmentation and oxidative stress [37]. Therefore, 12 days of micronutrient supplementation may have reduced sperm intracellular ROS concentrations and associated oxidative DNA damage of high fat diet fed males in this study via increased transfer of antioxidant factors into the lumen of the epididymis.

In addition to the transfer of antioxidants into the epididymal lumen the epididymis also makes antioxidants and antioxidant enzymes including superoxide dismutase (SOD) and glutathione (GSH). SOD an antioxidant enzyme is present in the caudal epididymis and protects sperm from ROS through dismutation of reactive oxygen to hydrogen peroxide, which is then rapidly converted to water [64,65]. While GSH is one of the only few antioxidants made by the male reproductive system, is highly expressed in the epididymis [66]. Zinc, one of the micronutrients included in this study, is a cofactor for SOD [64]. It has been shown that three months of zinc supplementation to sub-fertile men reduces sperm oxidative stress, DNA damage, and levels of apoptosis [67]. While, the polyphenols found in green tea can also up-regulate the expression of b-glutamylcystine synthetase, the rate-limiting enzyme in the synthesis of GSH. Studies in rodents have shown that levels of both SOD and GSH are increased in sperm, and concentrations of sperm lipid peroxidation and ROS are reduced after consumption of green tea extract [68]. In addition, green tea contains a number polyphenols (epicatechin, epicatechin-3-gallate, epigallocatechin, epigallocatechin-3-gallate, catechin and gallocatechin), that can bind lipid peroxides and/or interfere with production of lipid peroxidation [68] a hallmark of oxidative damage in sperm [24]. Therefore, our short-term micronutrient supplementation to high fat fed males could have also reduced sperm intracellular ROS concentrations and associated oxidative DNA damage by providing increased concentrations of cofactors required for the making and activity of epididymal antioxidants and antioxidant enzymes.

This is the first study to demonstrate that an oxidized base, namely 8OHdG that originates in mature sperm from male mice fed a high fat diet can persist in the paternal pronucleus post fertilization. With micronutrient supplementation of males fed a high fat diet able to restore 8OHdG levels in the male pronucleus back to control levels. In humans, sperm DNA integrity is important for successful fertilization and normal embryonic development, as evidenced by sperm with poor DNA integrity generating less successful pregnancies [69,70,71,72,73]. If the sperm chromatin is damaged (i.e., 8OHdG lesions) the oocyte is only equipped with limited machinery to repair this damage (such as OGG1), however a significant degree of DNA damage can result in delayed first and second cleavage events and blastocyst development via delays to paternal genome replication, presumably to allow for DNA repair prior to replication [51,74,75]. This is consistent with our study of increased 2-cell cleavage rates from our high fat diet fed males with micronutrient supplementation and previously published work, of the association of male obesity with failed fertilization and delayed 1st, 2nd, and 3rd cleavage events [10,13,76,77]. Likely, the improvements in cleavage rates of embryos sired by obese males fed micronutrient supplementation may be due to their reduced 8OHdG lesions in the male pronuclei. In addition, sperm function is also heavily influenced by nutrition, with a number of molecules including but not limited to zinc, vitamin D, vitamin A, epigallocatechin gallate (an active ingredient in green tea) being important for motility acquisition, capacitation, and fertilization [42,43,44,45] which could also explain the improvements to fertilization rates. 

A further finding of this study was that 12 days of micronutrient supplementation of obese males was able to partially restore fetal weights. Male obesity has been associated with a number of perturbed offspring phenotypes including increased fetal weights [9,11,12,13], increasing susceptibility of offspring to metabolic syndrome [8,9,14], sub-fertility [16], fatty liver disease [17], kidney disease [17], and hypertension [18]. Oxidative stress/high ROS concentrations in sperm are implicated as a causative factor for the transmission of paternal obesity to the next generation [78,79]. Artificial elevation of total sperm ROS concentrations by H_2_O_2_ prior to in vitro fertilization (IVF) recapitulated many of the offspring metabolic perturbations seen in their rodent models of paternal obesity, in which high concentrations of sperm ROS were noted [19]. In addition, more recently sperm small non coding RNAs (in particular sperm borne transfer RNAs (tRNAs)) and changes in abundance of sperm tRNAs and tRNA fragments have been implicated in the transmission of environmental health cues (including male obesity) to the early embryo and fetus [80,81,82,83]. Sperm tRNAs are of huge interest, given that tRNAs and tRNA fragments are scares in testicular sperm and only increase in abundance during the epididymal transit which can be modified via diet [81] and that in eukaryotes cleavage of 5’ and 3’ tRNA fragments are increased during oxidative stress [84]. Therefore, short-term micronutrient supplementation to obese males may be able to reduce the chronic disease risk in the next generation through directly reducing sperm ROS concentrations as well as reducing the oxidative environment of the epididymis thereby changing the abundance of sperm tRNAs delivered at fertilization, however further experiments are warranted. 

Similar to our previous studies we have shown that a reduction in adiposity is not required to improve sperm ROS concentrations because of male obesity [26], further confirming the concept that increased adiposity alone is not the cause of obesity-related subfertility. The duration of our treatment, 12 days, was designed to mimic a treatment that would be amenable to high degrees of compliance if adapted to a clinical setting or would allow males planning a pregnancy a short intervention period prior to conception. This duration is much shorter than all other studies (3–6 months) that have assessed the effects of micronutrient supplementation on male fertility [37] and for the treatment of other obesity-related conditions [85]. Therefore very short-term micronutrient supplementation in men who are obese may also illicit similar improvements to sperm quality, embryo quality, and fetal weights, however further studies are required.

## 5. Conclusions

Subfertility associated with male obesity may be restored with very short-term micronutrient supplementation that targets the transit of sperm through the epididymis, the time period during development sperm are most susceptible to oxidative damage. 

## Figures and Tables

**Figure 1 nutrients-11-02196-f001:**
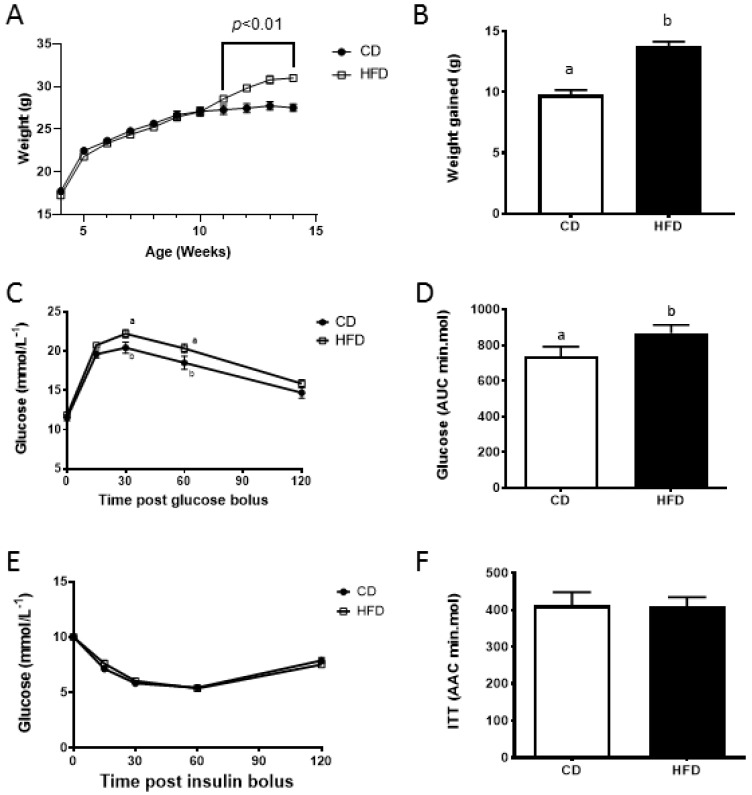
The effect of diet on body weight and glucose and insulin tolerance during initial diet phase. (**A**) Weekly weigh gain of males from 4 weeks until 14 weeks; (**B**) amount of total weight gained during the initial diet phase; (**C**) glucose tolerance as assessed by glucose tolerance test (GTT, 2g/kg) during initial diet phase; (**D**) glucose area under the curve (AUC, min.mmol) during GTT during initial diet phase; (**E**) insulin tolerance as assessed by insulin tolerance test (ITT, 1.0 IU) during initial diet phase; and (**F**) glucose area above the curve (AAC, min.mol) during ITT during initial diet phase. Data is expressed as mean ± standard error of the mean (SEM). *n* = 24 males per diet group. CD; control diet, HFD; high fat diet. Data was analyzed by a repeated measures analysis of variance (ANOVA) for weight gain or a one way ANOVA for GTT and ITT. Different letters denote significantly distinct groups at *p* < 0.05.

**Figure 2 nutrients-11-02196-f002:**
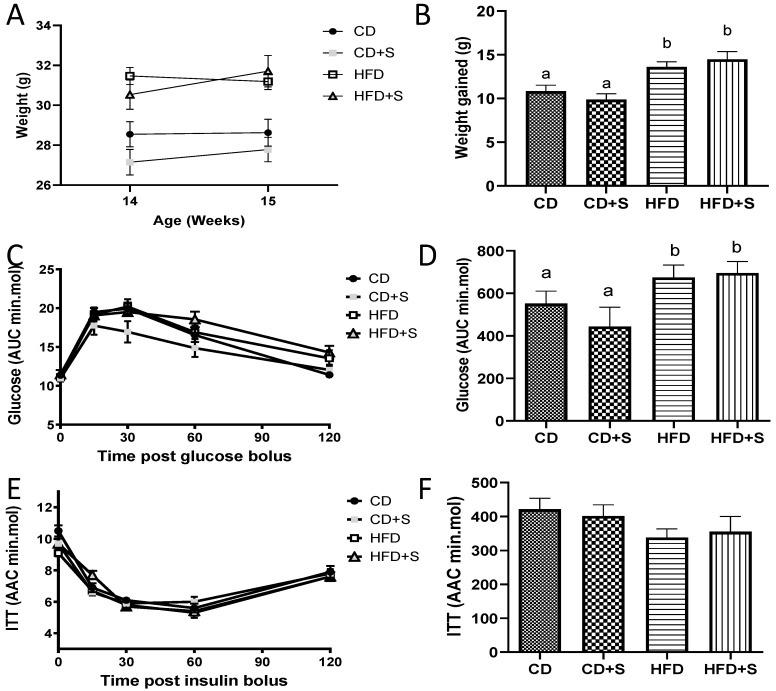
The effect of diet and short-term micronutrient supplementation on body weight and glucose and insulin tolerance during/after treatment phase. (**A**) Weekly weights between weeks 14 and 15; (**B**) total amount of weight gained during both the initial diet and treatment phase; (**C**) glucose tolerance as assessed by glucose tolerance test (GTT, 2g/kg) after treatment phase; (**D**) glucose area under the curve (AUC, min.mmol) during GTT after treatment phase; (**E**) insulin tolerance as assessed by insulin tolerance test (ITT, 1.0 IU) after treatment phase; and (**F**) glucose area above the curve (AAC, min.mol) during ITT after treatment phase. Data is expressed as mean ± SEM. *n* = 12 males per diet group. CD: control diet, CD + S: control diet plus micronutrient supplementation, HFD: high fat diet, HFD + S: high fat diet plus micronutrient supplementation. Data was analyzed by a repeated measures ANOVA for weight gain or a one way ANOVA for GTT and ITT. Different letters denote significantly distinct groups at *p* < 0.05.

**Figure 3 nutrients-11-02196-f003:**
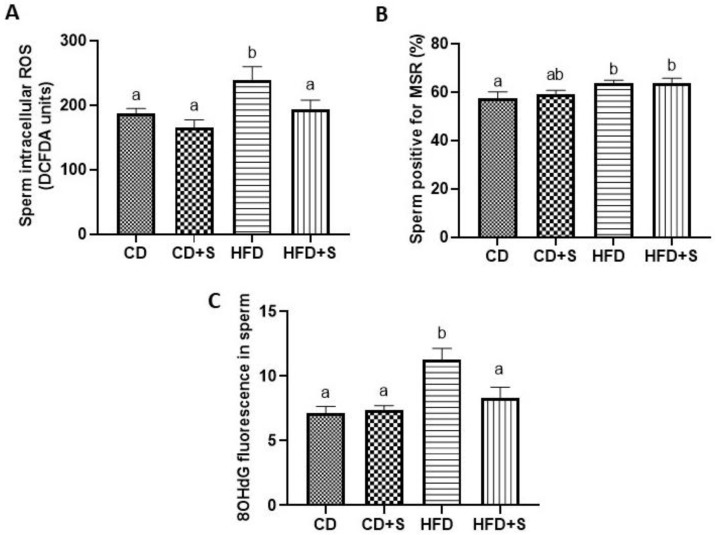
The effect of diet and short-term micronutrient supplementation on sperm ROS concentrations and oxidative DNA damage (8OHdG). (**A**) Sperm intracellular ROS concentrations (DCFDA), (**B**) sperm superoxide concentrations (MSR), and (**C**) sperm oxidative DNA damage (8OHdG). Data is expressed as mean ± SEM. *n* = 12 males per diet group. CD: control diet, CD + S: control diet plus micronutrient supplementation, HFD: high fat diet, HFD + S: high fat diet plus micronutrient supplementation. Data was analyzed by a one way ANOVA. Different letters denote significantly distinct groups at *p* < 0.05.

**Figure 4 nutrients-11-02196-f004:**
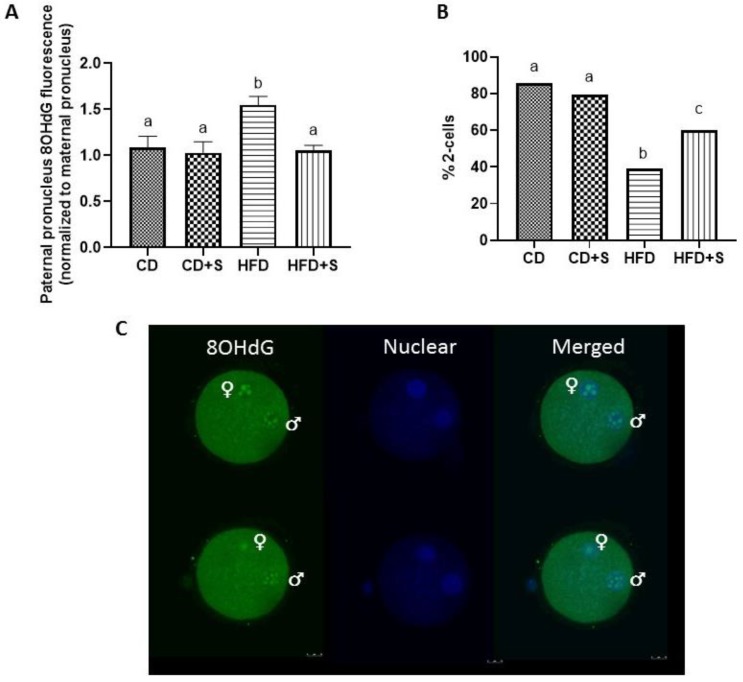
The effect of diet and short-term micronutrient supplementation on oxidative DNA damage (8OHdG) in the male pronucleus and 2-cell cleavage rates. (**A**) Quantified fluorescent intensity of the paternal pronucleus normalized to the maternal pronucleus in the subsequent embryo; (**B**) 2-cell cleavage rates; and (**C**) representative images of 8OHdG fluorescence in the male and female pronucleus of PN3 zygotes. Data are expressed as mean ± SEM. *n* ≥ 20 pronuclear embryos per diet group. Values are a proportion of total embryos for day 2 cleavage rates, *n* ≥ 100 embryos per diet group. CD: control diet, CD + S: control diet plus micronutrient supplementation, HFD: high fat diet, HFD + S: high fat diet plus micronutrient supplementation. 8ohdG in pronuclear embryos was analyzed by a one way ANOVA while 2-cell cleavage rates were analyzed by a binomial generalized linear model with LSD post hoc. Different letters denote significantly distinct groups at *p* < 0.05.

**Table 1 nutrients-11-02196-t001:** Composition of diets.

Diet		CD	CD + S	HFD	HFD + S
Main Ingredients					
Total fat (%)		6.0	6.0	21.0	21.0
Sucrose (%)		34.1	34.1	34.1	34.1
Wheat starch (%)		30.5	30.5	15.5	15.5
Vitamins and Minerals (micronutrients)					
Zinc (zinc sulphate monohydrate)		52 mg/kg	61 mg/kg	52 mg/kg	61 mg/kg
Selenium		0.3 mg/kg	0.44 mg/kg	0.3 mg/kg	0.44 mg/kg
Lycopene		-	0.3 mg/kg	-	0.3 mg/kg
Vitamin E (alpha-tocopherol acetate)		64 mg/kg	78 mg/kg	64 mg/kg	78 mg/kg
Vitamin C (ascorbic acid)		-	700 mg/kg	-	700 mg/kg
Folic acid		1 mg/kg	1.5 mg/kg	1 mg/kg	1.5 mg/kg
Green tea extract		-	0.95 mg/kg	-	0.95 mg/kg

CD: control diet, CD + S: control diet plus micronutrient supplementation, HFD: high fat diet, HFD + S: high fat diet plus micronutrient supplementation.

**Table 2 nutrients-11-02196-t002:** The effect of diet and short-term micronutrient supplementation on body composition at post mortem.

	CD (*n* = 12)	CD + S (*n* = 12)	HFD (*n* = 12)	HFD + S (*n* = 12)
Total body weight (g)	26.2 ± 0.6 ^a^	25.2 ± 0.5 ^a^	29.2 ± 0.5 ^b^	30.4 ± 0.8 ^b^
Adipose tissue (% of total body weight				
Peri-renal fat	0.21 ± 0.02 ^a^	0.20 ± 0.0 2 ^a^	0.32 ± 0.02 ^b^	0.37 ± 0.06 ^b^
Retroperitoneal fat	0.47 ± 0.03 ^a^	0.45 ± 0.03 ^a^	0.87 ± 0.06 ^b^	0.84 ± 0.08 ^b^
Omental fat	0.97 ± 0.07 ^a^	0.95 ± 0.06 ^a^	1.36 ± 0.08 ^b^	1.46 ± 0.05 ^b^
Dorsal fat	0.62 ± 0.03 ^a^	0.67 ± 0.04 ^a^	0.82 ± 0.04 ^b^	0.90 ± 0.04 ^b^
Gonadal fat	2.40 ± 0.12 ^a^	2.21 ± 0.13 ^a^	4.04 ± 0.19 ^b^	4.28 ± 0.19 ^b^
Sum of adipose tissues	4.66 ± 0.18 ^a^	4.49 ± 0.23 ^a^	7.41 ± 0.23 ^b^	7.86 ± 0.30 ^b^
Organs (g)				
Left testis	0.082 ± 0.005	0.085 ± 0.003	0.085 ± 0.006	0.085 ± 0.001
Right testis	0.082 ± 0.002	0.088 ± 0.001	0.150 ± 0.062	0.148 ± 0.063
Seminal vesicles	0.286 ± 0.013 ^a^	0.272 ± 0.010 ^a^	0.291 ± 0.006 ^a^	0.339 ± 0.009 ^b^
Liver	1.20 ± 0.05 ^ab^	1.11 ± 0.04 ^a^	1.19 ± 0.03 ^ab^	1.30 ± 0.07 ^b^
Pancreas	0.136 ± 0.008	0.135 ± 0.009	0.140 ± 0.008	0.149 ± 0.004
Left kidney	0.198 ± 0.008	0.183 ± 0.008	0.194 ± 0.005	0.189 ± 0.006
Right kidney	0.203 ± 0.011	0.190 ± 0.007	0.199 ± 0.006	0.207 ± 0.008
Metabolites and hormones				
Glucose (mmol/L^−1^)	10.6 ± 0.6	9.5 ± 0.3	10.7 ± 0.8	10.8 ± 0.7
Cholesterol (mmol/L^−1^)	3.1 ± 0.2 ^a^	3.0 ± 0.2 ^a^	4.2 ± 0.2 ^b^	4.3 ± 0.2 ^b^
HDL (mmol/L^−1^)	2.7 ± 0.2 ^a^	2.8 ± 0.1 ^a^	3.7 ± 0.02 ^b^	3.8 ± 0.3 ^b^
Triglycerides (mmol/L^−1^)	0.46 ± 0.04 ^a^	0.45 ± 0.03 ^a^	0.60 ± 0.05 ^b^	0.62 ± 0.06 ^b^
NEFA (mmol/L^−1^)	0.73 ± 0.03	0.72 ± 0.06	0.72 ± 0.03	0.77 ± 0.04
Testosterone (nmol/L^−1^)	0.05 ± 0.01	0.07 ± 0.01	0.05 ± 0.01	0.06 ± 0.01

Data is expressed as mean ± SEM. CD: control diet, CD + S: control diet plus micronutrient supplementation, HFD: high fat diet, HFD + S: high fat diet plus micronutrient supplementation. Data was analyzed by a one way ANOVA. Different letters denote significantly distinct groups at *p* < 0.05.

**Table 3 nutrients-11-02196-t003:** The effect of diet and short-term micronutrient supplementation on sperm motility, morphology, capacitation, and binding to the zona pellucida of MII oocyte.

	CD (*n* = 12)	CD + S (*n* = 12)	HFD (*n* = 12)	HFD + S (*n* = 12)
Sperm motility				
Progressive (%)	22.1 ± 2.9	16.0 ± 1.8	24.1 ± 3.9	20.4 ± 2.9
Non progressive (%)	42.3 ± 3.9	43.8 ± 3.2	36.8 ± 2.9	35.9 ± 3.5
Immotile (%)	35.5 ± 2.6	40.2 ± 4.3	39.1 ± 3.2	43.5 ± 3.2
Total motile (%)	64.5 ± 2.6	59.8 ± 4.3	60.9 ± 3.2	56.3 ± 3.2
Sperm morphology				
Normal (%)	53.5 ± 1.1 ^a^	58.5 ± 1.2 ^b^	45.1 ± 3.0 ^c^	47.6 ± 2.6 ^ac^
Head defect (%)	22.1 ± 1.1	21.6 ± 1.6	22.5 ± 2.2	20.7 ± 2.9
Tail and mid piece defect (%)	24.3 ± 0.6 ^a^	18.7 ± 1.3 ^b^	33.6 ± 1.4 ^c^	31.7 ± 1.0 ^c^
Sperm Capacitation				
Capacitated sperm (%)	90.6 ± 1.0 ^a^	89.5 ± 1.1 ^ab^	87.3 ± 0.9 ^b^	86.9 ± 0.6 ^b^
Non-capacitated sperm (%)	4.2 ± 0.6 ^a^	5.6 ± 0.6 ^ab^	5.3 ± 0.5 ^ab^	6.1 ± 0.4 ^b^
Acrosome reacted (%)	4.7 ± 0.4 ^a^	4.9 ± 0.6 ^a^	7.5 ± 0.6 ^b^	7.1 ± 0.6 ^b^
Sperm binding to the zona pellucida of MII oocyte				
Mean of sperm	20.4 ± 2.7	23.6 ± 3.5	22.1 ± 3.1	22.7 ± 3.1

Data is expressed as mean ± SEM. CD: control diet, CD + S: control diet plus micronutrient supplementation, HFD: high fat diet, HFD + S: high fat diet plus micronutrient supplementation. Data was analyzed by a one way ANOVA. Different letters denote significantly distinct groups at *p* < 0.05.

**Table 4 nutrients-11-02196-t004:** The effect of diet and short-term micronutrient supplementation on litter size and fetal growth.

	CD	CD + S	HFD	HFD + S
Litter size	8.8 ± 0.9	8.7 ± 0.7	8.9 ± 0.4	8.6 ± 0.5
Fetal weight (mg)	738.7 ± 23.4 ^a^	792.3 ± 22.9 ^ab^	805.4 ± 18.3 ^b^	783.5 ± 19.5 ^ab^
Crown rump length (mm)	17.4 ± 0.5	18.3 ± 0.6	18.6 ± 0.4	18.8 ± 0.5
Placenta weight (mg)	106.8 ± 4.1 ^a^	88.2 ± 4.1 ^b^	91.5 ± 3.2 ^b^	85.4 ± 3.4 ^b^
Fetal:placenta weight ratio	7.3 ± 0.4 ^a^	9.0 ± 0.4 ^b^	8.9 ± 0.3 ^b^	9.1 ± 0.3 ^b^

Data is expressed as mean ± SEM. Data is representative of 38 CD fetus, 25 CD + S fetus, 86 HFD fetus and 68 HFD + S fetus. CD: control diet, CD + S: control diet plus micronutrient supplementation, HFD: high fat diet, HFD + S: high fat diet plus micronutrient supplementation. Data were analyzed by a generalized linear model, with mother ID, father ID and litter size fitted as fixed effects and LSD post hoc test. Different letters denote significance at *p* < 0.05.

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
