# Peer review of "Dietary Micronutrient Supplementation for 12 Days in Obese Male Mice Restores Sperm Oxidative Stress"

_nutrients, 2019, doi:10.3390/nu11092196_

Round 1

Reviewer 1 Report

This is a nicely performed study showing that short-term dietary micronutrient supplementation can alleviate oxidative stress of sperm in obese mice, and contribute to improved embryo development and fetal growth. The data obtained here have further implications for sperm RNA biogenesis and their potential role for offspring health, I encourage the authors to discuss this interesting point as the authors have related research backgrounds. See my comments below.

It has been long known that ROS will increase the cleavage of tRNA and thus generate more tRNA-derived small RNAs (tsRNAs) or tRNA-derived fragments (tRFs) (RNA 2008, PMID: 18719243). This may provide a fundamental link to the biogenesis of sperm tsRNAs, which are known to contribute to regulate early embryo development and offspring phenotype (Science 2016 PMID: 26721685; Science 2016 PMID: 26721680; Nat Cell Biol 2018 PMID:29695786; PNAS 2019 PMID:31061112). Especially, it has been reported that the tsRNA composition is a predictor for sperm quality in regards to embryo developmental potential in human (Cell Discov 2019 PMID:30992999). These literature should be discussed in the context to further the finding of present work.

Author Response

Reviewer 1

This is a nicely performed study showing that short-term dietary micronutrient supplementation can alleviate oxidative stress of sperm in obese mice, and contribute to improved embryo development and fetal growth. The data obtained here have further implications for sperm RNA biogenesis and their potential role for offspring health, I encourage the authors to discuss this interesting point as the authors have related research backgrounds. See my comments below.

It has been long known that ROS will increase the cleavage of tRNA and thus generate more tRNA-derived small RNAs (tsRNAs) or tRNA-derived fragments (tRFs) (RNA 2008, PMID: 18719243). This may provide a fundamental link to the biogenesis of sperm tsRNAs, which are known to contribute to regulate early embryo development and offspring phenotype (Science 2016 PMID: 26721685; Science 2016 PMID: 26721680; Nat Cell Biol 2018 PMID:29695786; PNAS 2019 PMID:31061112). Especially, it has been reported that the tsRNA composition is a predictor for sperm quality in regards to embryo developmental potential in human (Cell Discov 2019 PMID:30992999). These literature should be discussed in the context to further the finding of present work.

Response:

We thank the reviewer for their kind words and agree that small non coding RNAs (in particular tRNAs) are likely implicated in paternal transmission of environmental cues. We have added a paragraph to the discussion to address their above point with particular focus on tRNAs and ROS in the context of our project.

(Pages 13-14 Lines 452-462)

Reviewer 2 Report

I have several remarks concerning this paper:

Figure 2: better resolution would be helpful. Also, labeling of individual groups in graphs A,C,E is not clear (especially Fig. 2C). It seems that the graph labeling of diet group CD and HFD+S is the same. Next, there is a difference between the graph column fill of diet group HFD in Fig. 2B and 2D,F.

Lines 248,249: Is there some difference in glucose intolerance of males fed a HFD, HFD+S vs. CD? (It is hard to see it in the graph)

Line 251: per-renal peri-renal

Line 259: Figure 2E,F link is missing.

Table 2.: The comments that the HFD diet increases right testis is missing.

Line 278: Title „Short term micronutrient supplementation improves aspects of sperm function“. This applies only to males fed a CD.

Line 285: Please, change the sentence: „…, with levels no different to males fed either a CD or HFD (Table 3). “ There is a difference between males fed a CD and HFD+S.

A better description of Figure 4C is needed (upper and lower panel).

Please, unify designation of 8OHdG e.g. Lines 342 and 427: 8ohdG, Line 398: 8OhdG

Line 345: Please, complete the title: the verb is missing.

Line 352: The information about increased fetal weight after micronutrient supplementation is missing.

Lines 357-359 should be deleted.

Author Response

Reviewer 2

Figure 2: better resolution would be helpful. Also, labelling of individual groups in graphs A,C,E is not clear (especially Fig. 2C). It seems that the graph labelling of diet group CD and HFD+S is the same. Next, there is a difference between the graph column fill of diet group HFD in Fig. 2B and 2D,F.

Response:

We have updated figure 2 to try an increase its resolution as well as characterisation of the individual groups within the graphs.

Lines 248,249: Is there some difference in glucose intolerance of males fed a HFD, HFD+S vs. CD? (It is hard to see it in the graph)

Response:

We have made it clearer in the results that addition of supplements to mice fed a HFD did not alter glucose tolerance compared with HFD, however it was still increased compared with CD males.

Lines 249-251

Line 251: per-renal → peri-renal

Response:

Has been updated.

Line 259: Figure 2E,F link is missing.

Response:

Figure 2E and F has been added.

Table 2: The comments that the HFD diet increases right testis is missing.

Response:

Although numerically increased the statistical analysis did not show that the HFD increased right testicular weights (please see large SEM).

Line 278: Title „Short term micronutrient supplementation improves aspects of sperm function“. This applies only to males fed a CD.

Response:

Line 278 has been updated to inform that it only applied to males fed a CD.

Line 285: Please, change the sentence: „…, with levels no different to males fed either a CD or HFD (Table 3). “ There is a difference between males fed a CD and HFD+S.

Response:

This line has been updated to inform the readers of the differences in sperm morphology in those mice fed a HFD irrespective of micronutrient supplementation particular sperm with tail defects compared with those males fed a CD and CD+S.

Lines 284-290

A better description of Figure 4C is needed (upper and lower panel).

Response:

We have reworded the description of Figure 4C to better identify the figure.

Please, unify designation of 8OHdG e.g. Lines 342 and 427: 8ohdG, Line 398: 8OhdG

Response:

These omissions have been fixed.

Line 345: Please, complete the title: the verb is missing.

Response:

The title has been fixed to replace the missing verb

Line 352: The information about increased fetal weight after micronutrient supplementation is missing.

Response:

Although numerically increased fetal weights between CD and CD plus micronutrient supplementation (CD+S) was not different and therefore this has not been mentioned in the results. 

Lines 357-359 should be deleted.

Response:

These lines have been deleted.